# Concomitant Cervical and Anal Screening for Human Papilloma Virus (HPV): Worth the Effort or a Waste of Time? [note 1]

**DOI:** 10.3390/cancers16203534

**Published:** 2024-10-19

**Authors:** Camille Chilou, Iolanda Espirito Santo, Seraina Faes, Pénélope St-Amour, Martine Jacot-Guillarmod, Basile Pache, Martin Hübner, Dieter Hahnloser, Fabian Grass

**Affiliations:** 1Department of Visceral Surgery, Lausanne University Hospital (CHUV), University of Lausanne (UNIL), 1011 Lausanne, Switzerland; camille.chilou@chuv.ch (C.C.);; 2Stadtspital Triemli Zürich, 8063 Zürich, Switzerland; 3Gynecology Department, Department Women-Mother-Child, Lausanne University Hospital (CHUV), 1011 Lausanne, Switzerland; martine.jacot-guillarmod@chuv.ch (M.J.-G.);

**Keywords:** HPV testing, papillomavirus, proctology, screening, anal squamous intraepithelial neoplasia, cancer

## Abstract

Anal human papilloma virus (HPV) is the leading etiology of anal and cervical cancer. While cervical HPV screening for women is well established, anal HPV screening is restrained to a limited population and specific indications. Our group performed concomitant anal and cervical screening in 275 women at a single gynecologic appointment and revealed high-risk anal HPV in 91 women (33%). The present follow-up study intended to analyze the outcomes of these women during a specialized follow-up in a dedicated high-resolution anoscopy (HRA) outpatient clinic. Independent risk factors for anal HPV were anal intercourse and the presence of cervical HR-HPV with a three- and fourfold increased risk, respectively.

## 1. Introduction

Human Papillomavirus (HPV) is the most common sexually transmitted disease, with an estimated prevalence of 70–80% in women [1]. Furthermore, high-risk (HR) HPV association is seen in 90% of cervical and anal cancers [2]. The pathophysiology of anal and cervical carcinoma, secondary to HPV infection, is quite similar. The cells make contact with the virus when a breach in the cutaneous or mucosal epithelial barrier occurs, leading to exposure [3,4]. Infection spreads with the division of infected cells leading to a wide variety of lesions, influenced by the failure of control mechanisms, including intra-cellular control, a paracrine signaling cascade, and diminished immunological control [5]. Nearly 200 subtypes of HPV have been reported. Among them, 15 predispose people to an increased risk of squamous intraepithelial lesions (SIL) and squamous cell carcinoma [6]. In 90% of cases, the infection is transient and spontaneously resolves [7]. However, high-grade squamous intraepithelial lesions (H-SIL) are unlikely to spontaneously regress [8]. Several risk factors for disease progression, including multiple HPV infections, anal intercourse, immune suppression, human immunodeficiency virus (HIV), smoking, and social deprivation, have been described [9]. While HPV infection has been extensively studied in women with HIV [10], only scarce data are available on general prevalence and risk factors of anal SIL in women [11,12]. Switzerland has been identified as a country with one of the highest age-standardized incidence rates of anal cancer in women [13]. Furthermore, recent data shows an increase on anal cancer incidence in the younger, sexually active population [14].

Our group conducted a prospective study on concomitant cervical and anal screening in a tertiary academic institution to determine the risk of combined infections [15]. This follow-up study aimed to analyze the outcomes of women with an abnormal finding on anal smears or biopsies at initial screening who were surveilled in a specialized high-resolution anoscopy (HRA) clinic.

## 2. Materials and Methods

### 2.1. Study Settings, Participants, and Sampling

This is a follow-up study of the *AnusGynecology* (ANGY) prospective cross-sectional single-center study (SNCTP000002567) [15]. In total, 275 women were recruited at the Lausanne University Hospital (CHUV) in Switzerland. Details about recruitment and evaluations were described in the original study protocol [16]. In brief, patients initially presented to a single screening appointment at the gynecologic outpatient clinic. Detailed history, clinical examination, colposcopy, and HRA with concomitant cervical and anal HPV screening was performed by board-certified gynecologists and colorectal surgeons, respectively. During the same appointment, anal and cervical smears for cytology were obtained. Suspicious condyloma-like lesions were biopsied and sent to the institutional pathology lab for further analysis. HPV tests were performed with the MagNaPure 96 Total Nucleic Acids kit (Roche, Basel, Switzerland), with genotyping of 28 HPV genotypes (6, 11, 16, 18, 26, 31, 33, 35, 39, 40, 42–45, 51–54, 56, 58, 59, 61, 66, 68–70, 73, and 82) by the Anyplex™ II HPV28 kit (Seegene, Seoul, Republic of Korea). To establish individual risk profiles, HIV testing was systematically performed and a self-assessment questionnaire accounting for sexual habits and history was distributed to all patients.

For the purpose of this present study, patients were divided into two groups according to their anal HR-HPV status (normal vs. pathologic finding at initial screening). Baseline demographic data, social habits including smoking and sexual history (type of intercourse, number of sexual partners, use of condom, HPV vaccination status), previous anorectal disease or surgery, and cervical HR-HPV status were assessed.

All women with positive HR-HPV (predominantly types 16 and 18) at anal smears during the proctologic exam at the initial ANGY screening visit constituted the cohort for this present study. A follow-up visit at 12 weeks at the specialized HRA outpatient clinic was scheduled for all these women. Our institution offers a specialized weekly HRA consultation led by a board-certified colorectal surgeon, totaling around 300 outpatient visits per year [17]. Most referrals derive from HIV infectious disease specialists (43%), gastroenterologists, gynecologists, and general practitioners. All referred ANGY patients underwent detailed clinical examination including HRA screening, anal smears, and anal biopsies, if indicated. Smears were collected using a plastic swab in a spiral motion movement. Excisional biopsy of suspicious lesions was performed under local anesthesia. Otherwise, cryosurgery was performed in 3 two-weekly intervals. More extensive lesions were excised in the institutional outpatient surgery center in an ambulatory setting. Alternatively, CO_2_ laser therapy and topical agents (imiquimod, 5-fluorouracile) were applied. Histopathological examination of the specimens was carried out by the institutional Department of Pathology.

### 2.2. Statistical Analysis

Descriptive statistics for categorical variables were reported as frequencies (percentages) and continuous variables were reported as means (SD). The χ^2^ test was used to compare categorical variables. All the statistical tests were 2-sided, and a level of 0.05 was used to indicate statistical significance. Significant and clinically relevant variables were entered into a multivariable logistic regression (based on a probit regression) model to provide adjusted estimations of the odds ratio (OR) with 95% confidence intervals (CIs). Data analysis was performed with the SPSS Advanced Statistics 29 (IBM Software Group, Chicago, IL, USA) and GraphPad Prism Version 10.1.2.

## 3. Results

Among the 275 women enrolled in the initial study (age 42 ± 12 years), 91 (33%) tested positive for anal HR-HPV at the initial screening visit. Table 1 summarizes the demographics and sexual histories of the cohort.

Of the 91 participants, 48 (53%) presented to the specialized HRA outpatient screening visit 12 weeks later. Figure 1 summarizes the findings of these women during initial screening in the setting of the ANGY study and during specialized HRA follow-up.

Among the 48 women presenting to the HRA follow-up visit, 31 (65%) presented with persistent anal HR-HPV, while biopsies revealed anal SIL of any type in 16 women (33%). Of the 32 women with a normal exam during HRA exams, 60% were anal carriers of HR-HPV subtypes.

Most women (24/31, 77%) attended at least a second follow-up visit, which was suggested to all women with positive anal HR-HPV. The median length of follow-up was 42 months (1–56). During the follow-up, 2 patients developed L-SIL and 3 evolved to H-SIL dysplastic lesions.

Figure 2 displays the results of the multivariable analysis, revealing cervical high-risk HPV infection (OR 4, 95% CI 2.2–7.5) and anal intercourse (OR 3.1, 95% CI 1.7–5.9) as independent risk factors for anal HR-HPV infection (both *p* < 0.001).

As a result of these findings, the institutional treatment and surveillance algorithm was adapted, as demonstrated in Figure 3.

## 4. Discussion

The present study revealed a fourfold increased risk of concomitant anal HR-HPV infection in women with cervical HR-HPV, which is further magnified by sexual behaviors such as anal intercourse and, to a lesser extent, promiscuity. The study further demonstrated reluctance to specialized follow-up with only half of women with a finding of anal HR-HPV at initial screening presenting to specialized HRA follow-up 12 weeks later. Since the initial finding of anal HR-HPV was confirmed at the follow-up visit in two-thirds of these women, with detection of dysplastic lesions in 30% and disease progression in at least some patients, follow-up in a specialized setting appears mandatory. The findings of this study emphasize the need for proper screening and patient counseling during gynecologic screening visits to raise awareness of exposure, discuss behavioral risk factors, and mitigate the risk of disease progression to cancer.

The finding of 33% concomitant HPV infection in the present series compares well to former reports [18]. A similarly powered retrospective study on 272 Czech women found concurrent infections in 42% with cervical HR-HPV, with anal intercourse representing the major risk factor [19]. This finding was confirmed by a systematic review of 23 publications, notably highlighting an increased risk in HIV-positive women and women with a history of HPV-related lower genital tract pathology [20]. A population-based large-scale study from Denmark revealed a hazard ratio for anal dysplastic lesions of 2.9 for women with HR-HPV compared to women testing HPV-negative [21]. Importantly, the same study revealed a twofold increased risk of anal squamous cell carcinoma in HR-HPV-positive women, further supporting the urgent need for close anal surveillance in this population. Of particular interest—given the similar size, setting, and patient characteristics—is a recently published Chinese cohort study [22]. Anal cytological abnormalities were found in 41% of the cohort; however, the rate of anal HR-HPV was as high as 71% in women with cervical HR-HPV, with HPV type 16 being the most prevalent subtype. Similar to the present series, a three–fourfold increased risk of anal HR-HPV infection was detected, with immunosuppression being identified as a major risk factor.

The results of the present study warrant an adapted surveillance protocol in our institution and help to answer frequently and repeatedly asked questions of surveilled women in our institution. Anal swabs should be routinely performed in women with cervical HR-HPV. Furthermore, these women need to be aware of the increased risk of anal SIL and cancer, warranting surveillance in a dedicated HRA clinic. The suggested treatment algorithm (Figure 3) is intended to help with guidance in clinical practice.

Prevention is key in the setting of HPV-induced cancer. A recently published common consensus statement of several European societies on the management of vaginal intraepithelial neoplasia follows the principles for prevention of SIN at other anogenital sites and presumes avoiding risk factors such as smoking, long-term oral contraception, multiple sexual partners, and unsafe sex [23]. Furthermore, the guideline suggests particular caution with immunosuppressed, HIV-positive, and transplant patients, and highlights the importance of education about regular follow-up. The implementation of vaccination is expected to further contribute to the prevention of vaginal, cervical, and anal intraepithelial neoplasia [24,25].

Detailed information appears to be key, given the low compliance to the suggested surveillance protocol, with only half of women presenting to the follow-up visit. Many women may be reluctant to regular proctology follow-up, as seen in a recently published nationwide registry study from Denmark [26]. However, follow-up is crucial given the persistence and even progression of the conditions during the short follow-up period of this present series, a finding which has not been specifically investigated so far. In most countries, anal screening is mainly performed when a suspicious lesion is visualized, or cervical cancer is diagnosed [27]. However, anal cytology and HPV screening are simple to perform, painless, and can be easily performed during any gynecologic exam, providing logistic prerequisites are met. Moreover, anal smears for HPV testing have a specificity and sensitivity of 42% and 92%, respectively, according to a recent meta-analysis on HPV-related biomarkers for anal cancer screening [28].

HIV infection has been identified as major risk factor in former studies and has been extensively studied [29]. As part of the protocol of the present study, testing was performed systematically; however, only one patient newly tested positive in this series. Hence, this evident risk factor was not included in the risk factor analysis. On the other hand, anal intercourse and, to a lesser degree, intercourse with several sexual partners, increase the risk of exposure to HPV infection of the anal region. The follow-up visit fosters a thorough discussion regarding sexual behavior, proper contraception, and safety measures to raise awareness and for primary prevention purposes. At the same occasion, the value and impact of HPV vaccination should be discussed, given its safety and efficacy in the non-HIV population [30]. Importantly, HPV vaccination at a younger age is associated with substantially reduced risk of anal high-grade SIL or (pre-)cancerous lesions in the general population [31].

Several limitations of this study need to be considered. First, the cohort is heterogeneous due to the focus on all-comers without specific risk profiles. Some of the risk factors were underrepresented and impeded further analysis of the rather small cohort. Second, follow-up was limited with a rather short surveillance period and almost 50% of patients who did not present to the scheduled visits. This may potentially have induced selection bias. Future larger studies may be able to provide more insights on the impact of different genotypes and viral load on the development of dysplastic lesions and anal cancer.

## 5. Conclusions

Concomitant cervical and anal HPV infection is common, and the prevalence of HR-HPV and consecutive SIL is high in sexually active women. Anal screening of women with cervical HR-HPV infection hence appears to be worth the effort; follow-up by colorectal specialists is strongly recommended, given the substantially increased risk of anal HR-HPV infection and the potential of disease progression to anal cancer. Proper patient counseling during gynecologic screening visits is mandatory to increase awareness and compliance to follow-up.

## Figures and Tables

**Figure 1 cancers-16-03534-f001:**
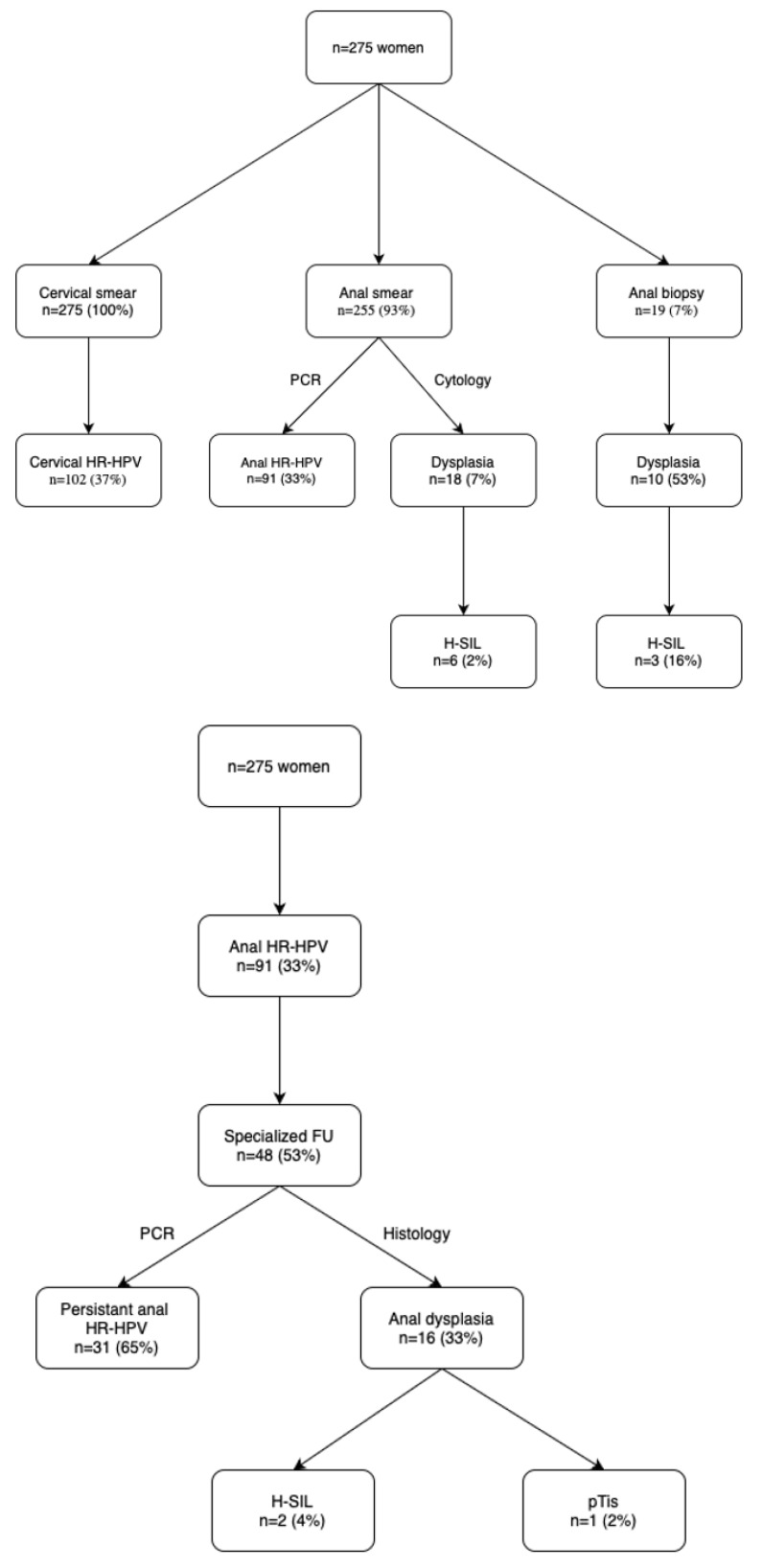
Findings during screening visits. Displayed are initial findings (during ANGY study, top chart) and during specialized HRA follow-up (bottom chart). HR-HPV—high-risk HPV; H-SIL—high-risk squamous intraepithelial lesion; FU—follow-up; pTis—carcinoma in situ.

**Figure 2 cancers-16-03534-f002:**
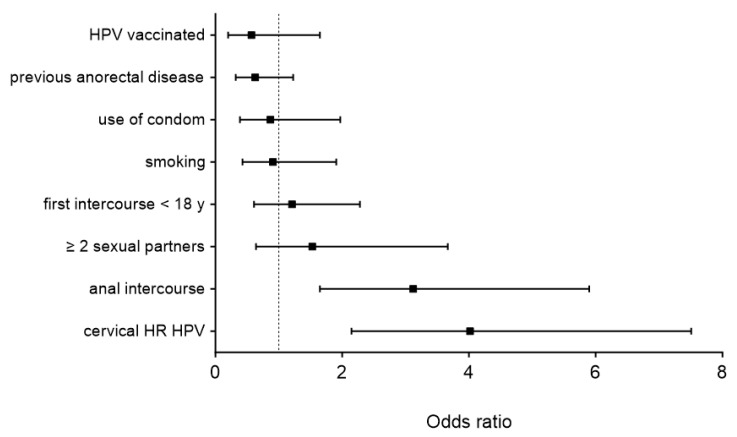
Multivariate analysis of risk factors associated with anal high-risk HPV. HPV—human papilloma virus; y—years; HR—high-risk. Displayed are odds ratios (squares) and 95% confidence intervals (brackets).

**Figure 3 cancers-16-03534-f003:**
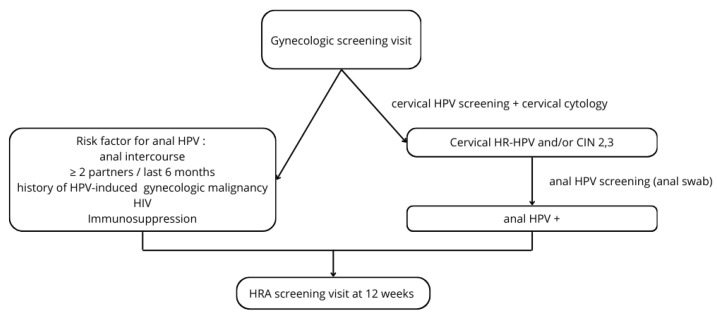
Treatment algorithm for women at risk. HPV—human papilloma virus; HIV—human immunodeficiency virus; HR—high-risk; CIN—cervical intraepithelial neoplasia.

**Table 1 cancers-16-03534-t001:** Demographics of the study cohort.

Item	All Patients (n = 275)	Anal HR-HPV (n = 91)	Controls (n = 184)	*p*
Age (years, mean ± SD)	42 ± 12	41 ± 12	43 ± 12	0.141
Smoking (%)	62/263 (24)	24/89 (27)	38/174 (22)	0.361
Cervical HR-HPV (%)	102 (37)	54 (59)	48 (32)	<0.001
≥2 sexual partners (%)	208/265 (78)	79/88 (90)	129/177 (73)	0.001
Age of first intercourse < 18 y (%)	134/262 (51)	43/88 (49)	91/174 (52)	0.604
Use of condom (%)	49/266 (18)	16 (18)	33/175 (19)	0.869
Anal intercourse (%)	97/270 (36)	48 (53)	49/179 (27)	<0.001
Previous anorectal disease (%)	84/267 (31)	26/89 (29)	58/178 (33)	0.675
HPV vaccinated (%)	25/261 (10)	9/88 (10)	16/173 (9)	0.826

SD—standard deviation; HR-HPV—high-risk HPV; y—years; HPV—human papillomavirus.

## Data Availability

The data presented in this study are available on request from the corresponding author. The data are not publicly available due to ethical comity restrictions.

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
