# Peer review of "Concomitant Cervical and Anal Screening for Human Papilloma Virus (HPV): Worth the Effort or a Waste of Time?†"

_cancers, 2024, doi:10.3390/cancers16203534_

Round 1

Reviewer 1 Report

Comments and Suggestions for Authors

The article "Concomitant cervical and anal screening for human papilloma virus (HPV): Worth the effort or waste of time?" is the result of long waited ANGY study. The protocol for this excellent cross-sectional prospective clinical study was published in 2022 gives all methodological details. The results justify the expectations and will be a significant contribution to the rising issue of anal intraepithelial neoplasia in women.

The article is clearly written, understandable and provides all data planned by the design of ANGY study.

The only suggestion I have is to avoid the use of term Dysplasia. It is not compatible with the use of LAST terminology which the whole article refers to (i.e. LSIL and HSIL lesions of the cervix and anus). Please, correct term dysplasia with squamous intraepithelial neoplasia (SIL).

Author Response

Comments 1: The article "Concomitant cervical and anal screening for human papilloma virus (HPV): Worth the effort or waste of time?" is the result of long waited ANGY study. The protocol for this excellent cross-sectional prospective clinical study was published in 2022 gives all methodological details. The results justify the expectations and will be a significant contribution to the rising issue of anal intraepithelial neoplasia in women. The article is clearly written, understandable and provides all data planned by the design of ANGY study.

Reply 1: We would like to thank the reviewer for the encrouaging feedback. Indeed, we are concerneda about underreported intraepthelial neoplasia in women. Even though we report on a rather small cohort, we believe the study helps to rise the awareness in our population and contributes to the increasing body of evidence regarding HPV infection in sexually active women.

Comment 2: The only suggestion I have is to avoid the use of term Dysplasia. It is not compatible with the use of LAST terminology which the whole article refers to (i.e. LSIL and HSIL lesions of the cervix and anus). Please, correct term dysplasia with squamous intraepithelial neoplasia (SIL).

Thank you for this comment. We corrected the terminology throughout the manuscript accordingly.

Reviewer 2 Report

Comments and Suggestions for Authors

This article is really interesting because it brings to light a topic that is quite forgotten, at least in our environment. The article is well structured and explained and the results are clear.

There are only two suggestions I would like to make: in material and method, it is not explained at any point what method is used for the detection of HPV and for its genotyping and it would be important to clarify this, in order to know the sensitivity and specifity. On the other hand, in my opinion, figure 3 should be in the results section not in discussion.

As a further suggestion, it would be very interesting to know in future studies more data on the other genotypes involved in anal cancer, as well as whether the amount of viral load influences the possible development of dysplastic lesions.

In my opinion, it is a very interesting article and of interest to many different types of specialists.

Author Response

Comment 1: This article is really interesting because it brings to light a topic that is quite forgotten, at least in our environment. The article is well structured and explained and the results are clear.

Reply 1: We would like to thank reviewer 2 for the encouraging feedback. We agree that the issue of concomitant HPV infection is underreported, maybe even neglected to some degree at least in our clinical practice. We hope that our study helps to increase awareness among clinicians and patients likewise.

Comment 2: There are only two suggestions I would like to make: in material and method, it is not explained at any point what method is used for the detection of HPV and for its genotyping and it would be important to clarify this, in order to know the sensitivity and specifity. On the other hand, in my opinion, figure 3 should be in the results section not in discussion.

Reply 2: We added this important information to the methods section including which kit was used. We also introduced Figure 3 in the results section as suggested.

Comment 3: As a further suggestion, it would be very interesting to know in future studies more data on the other genotypes involved in anal cancer, as well as whether the amount of viral load influences the possible development of dysplastic lesions.

Reply 3: Thank you for this thoughtful suggestion, we implemented these suggestions at the end of the manuscript.

Reviewer 3 Report

Comments and Suggestions for Authors

Dear authors,

thank you for this interesting paper on concomitant anal and cervical hpv infection

however, i would not support publishing it.

Introduction is really short and lacks scientific rigor, has many flaws and is not readable.

methods are fine

results are very weak and give little new information to published literature

conclusions are based on little and already known data

Comments on the Quality of English Language

extensive

Author Response

Comment 1: Dear authors, thank you for this interesting paper on concomitant anal and cervical hpv infection however, i would not support publishing it. Introduction is really short and lacks scientific rigor, has many flaws and is not readable.

Reply 1: We would like to thank the reviewer for the input and time to revise our manuscript. We tried to add  information to the introduction section in order to provide a more extensive paragraph to situate the research question. We kindly remind the reviewer that for a comment, the word count is limited. Hence, we tried to be as concise as possible. 

Comment 2: methods are fine

Reply 2: Thank you for this comment.

Comment 3: Results are very weak and give little new information to published literature. Conclusions are based on little and already known data.

Reply 3: While we agree that our study has limitations related to a rather small sample size and low compliance to the suggested follow-up, we believe that the study adds to the existing literature. First, we provide "real life data" on unselected all- comers. Second, several findings of the study are of interest: The non willingness of a substantial proportion of women to consider specialized follow up in the HRA clinic, the significant number of patients with (underreported) anal HPV-related dysplasia, and the multivariable analysis revealing an (independent) 4-fold increased risk of anal HR HPV infection in patients with cervical HR HPV infection. We are glad to quantify the risk in our population and daily clinical practice and adapted our algorithm accordingly. We also discuss obvious limitations of our study.

On a personal note, I can confirm that our group has extensive experience in scientific writing and proficiency in English writing. We try our best to revise / edit English where needed but are convinced that minor editing as offered by the EO will be sufficient to correct potential residual minor issues.

Again, we would like to thank the reviewer for taking the time to comment on our study.

Round 2

Reviewer 3 Report

Comments and Suggestions for Authors

Dear authors, the manuscript has been improved and it may deserve consideration for publication. 

I would still enlarge the scope of the preventive effort, also to those affected by vaginal intraepithelial neoplasia (10.1097/LGT.0000000000000732)

Thank you

Comments on the Quality of English Language

Minor

Author Response

Comment 1: 

Dear authors, the manuscript has been improved and it may deserve consideration for publication. 

I would still enlarge the scope of the preventive effort, also to those affected by vaginal intraepithelial neoplasia (10.1097/LGT.0000000000000732)

Reply 1: We would like to thank the reviewer for the reevaluation of our manuscript and the encouraging comment. We gladly expanded on the prevention strategy on vaginal intraepithelial nepolasia by adding a dedicated paragraph in the discussion section (lines 180-188).

Again, we thank the reviewer for time and effort to help improve our manuscript.

Round 3

Reviewer 3 Report

Comments and Suggestions for Authors

The  manuscript has been improved and now deserves publication.